# Sales of antibiotics and hydroxychloroquine in India during the COVID-19 epidemic: An interrupted time series analysis

Giorgia Sulis[1,2], Brice Batomen[3], Anita Kotwani[4], Madhukar Pai[1,2], Sumanth Gandra[5]*

1 Department of Epidemiology, Biostatistics and Occupational Health, McGill University, Montreal, Quebec, Canada, 2 McGill International TB Centre, McGill University, Montreal, Quebec, Canada, 3 Dalla Lana School of Public Health, University of Toronto, Toronto, Ontario, Canada, 4 Department of Pharmacology, Vallabhbhai Patel Chest Institute, University of Delhi, Delhi, India, 5 Division of Infectious Diseases, Department of Medicine, Washington University in St. Louis, Saint Louis, Missouri, United States of America

* gandras@wustl.edu

**Data Availability Statement:** Data cannot shared publicly because of license agreement with the IQVIA Inc. The data underlying the results

## Abstract

### Background

We assessed the impact of the coronavirus disease 2019 (COVID-19) epidemic in India on the consumption of antibiotics and hydroxychloroquine (HCQ) in the private sector in 2020 compared to the expected level of use had the epidemic not occurred.

### Methods and findings

We performed interrupted time series (ITS) analyses of sales volumes reported in standard units (i.e., doses), collected at regular monthly intervals from January 2018 to December 2020 and obtained from IQVIA, India. As children are less prone to develop symptomatic severe acute respiratory syndrome coronavirus 2 (SARS-CoV-2) infection, we hypothesized a predominant increase in non-child-appropriate formulation (non-CAF) sales. COVID-19-attributable changes in the level and trend of monthly sales of total antibiotics, azithromycin, and HCQ were estimated, accounting for seasonality and lockdown period where appropriate. A total of 16,290 million doses of antibiotics were sold in India in 2020, which is slightly less than the amount in 2018 and 2019. However, the proportion of non-CAF antibiotics increased from 72.5% (95% CI: 71.8% to 73.1%) in 2019 to 76.8% (95% CI: 76.2% to 77.5%) in 2020. Our ITS analyses estimated that COVID-19 likely contributed to 216.4 million (95% CI: 68.0 to 364.8 million; $P = 0.008$) excess doses of non-CAF antibiotics and 38.0 million (95% CI: 26.4 to 49.2 million; $P < 0.001$) excess doses of non-CAF azithromycin (equivalent to a minimum of 6.2 million azithromycin treatment courses) between June and September 2020, i.e., until the peak of the first epidemic wave, after which a negative change in trend was identified. In March 2020, we estimated a COVID-19-attributable change in level of +11.1 million doses (95% CI: 9.2 to 13.0 million; $P < 0.001$) for HCQ sales, whereas a weak negative change in monthly trend was found for this drug. Study limitations include the lack of coverage of the public healthcare sector, the inability to distinguish antibiotic and HCQ sales in inpatient versus outpatient care, and the suboptimal number of pre-

presented in the study are available from IQVIA Consulting and Information Services India Pvt. Ltd. https://www.iqvia.com/locations/india.

**Funding:** The authors received no specific funding for this work.

**Competing interests:** I have read the journal's policy and the authors of this manuscript have the following competing interests: MP is a member of the Editorial Board of PLOS Medicine, he coedits the PLOS Tuberculosis Channel and is the Editor-in-Chief of PLoS Global Public Health. All the other authors have no conflicts of interest to declare.

**Abbreviations:** ATC, Anatomical Therapeutic Chemical; AWaRe, Access, Watch, Reserve; CAF, child-appropriate formulation COVID-19, coronavirus disease 2019; ESBL, extended spectrum beta-lactamase; HCQ, hydroxychloroquine; ITS, interrupted time series; *S.* Typhi, *Salmonella enterica* serotype Typhi; SARS-CoV-2, severe acute respiratory syndrome coronavirus 2.

and post-epidemic data points, which could have prevented an accurate adjustment for seasonal trends despite the robustness of our statistical approaches.

## Conclusions

A significant increase in non-CAF antibiotic sales, and particularly azithromycin, occurred during the peak phase of the first COVID-19 epidemic wave in India, indicating the need for urgent antibiotic stewardship measures.

## Author summary

### Why was this study done?

- There are concerns that the widespread and often inappropriate use of antibiotics has been aggravated by the COVID-19 pandemic, but little is known regarding the true impact of the pandemic on antibiotic use, particularly in low- and middle-income countries (LMICs).

- India is the largest antibiotic user in the world and is among the countries that are most severely affected by the pandemic.

- About 75% of healthcare in India is private, and this unregulated and fragmented private sector accounts for 90% of antibiotic consumption, raising major concerns about the potential effects of COVID-19 on prescribing and dispensing practices.

### What did the researchers do and find?

- Using an interrupted time series (ITS) design, we examined sales volumes of total antibiotics, azithromycin alone, and hydroxychloroquine (HCQ) in India's private sector from January 2018 to December 2020.

- Focusing on non-pediatric formulations and adjusting for underlying seasonal and non-seasonal trends and accounting for the effect of lockdown, we estimated the impact of the first epidemic wave on monthly sales.

- Based on our models, COVID-19 likely contributed to about 216 million excess doses (95% CI: 68.0 to 364.8 million; $P = 0.008$) of total antibiotics and 38.0 million excess doses (95% CI: 26.4 to 49.2 million; $P < 0.001$) of azithromycin between June and September 2020 (i.e., after the lockdown and until the epidemic peak).

- HCQ sales peaked in March 2020, reflecting the widespread use of this drug for both prophylaxis and treatment of COVID-19 (+11.1 million doses [95% CI: 9.2 to 13.0 million]; $P < 0.001$), followed by a slow decline afterwards.

### What do these findings mean?

- Our findings indicate a significant increase in antibiotic sales, particularly of azithromycin, during the peak phase of the first COVID-19 epidemic wave in India.

- Similar trends are likely to have occurred in other LMICs, where antibiotics are often overused.

- The medium- and long-term consequences for bacterial resistance patterns are highly concerning, highlighting the need for urgent antibiotic stewardship measures.

## Introduction

India is the largest consumer of antibiotics in the world [1,2]. Broad spectrum antibiotics such as second- and third-generation cephalosporins, macrolides, and quinolones are overused for acute respiratory tract infections in India [3]. There is a concern that symptomatic severe acute respiratory syndrome coronavirus 2 (SARS-CoV-2) infection, which causes coronavirus disease 2019 (COVID-19), could lead to a substantial increase in antibiotic consumption (often inappropriately), thus promoting antibiotic resistance [4].

In many countries, azithromycin and hydroxychloroquine (HCQ) are reportedly being used off label in prophylactic and therapeutic regimens either alone or in combination. In India, azithromycin is typically utilized to treat a range of conditions, including acute respiratory tract infections, bacterial dysentery, and enteric fever [5]. This macrolide antibiotic was repurposed for the treatment of COVID-19 based on in its hypothetical anti-inflammatory and immunomodulatory properties [6–8]. On the other hand, HCQ in India is mainly utilized for treatment of autoimmune diseases, such as rheumatoid arthritis and systemic lupus erythematosus, and post-viral infectious arthritis, such as chikungunya arthritis, and is not part of national malaria treatment guidelines [9–12]. It has been suggested that HCQ could have antiviral activity as well as indirect anti-inflammatory properties through the activation of CD8+ T cells and the reduction of pro-inflammatory cytokine response, thus leading to its widespread use in the management of COVID-19 as well as in pre- and post-exposure prophylaxis [13,14]. However, an increasing number of studies have observed no beneficial effects from the use of azithromycin and/or HCQ, and a number of safety concerns have also been raised [15–19].

A growing body of evidence from observational studies across multiple countries consistently indicates that only a small proportion of hospitalized COVID-19 patients develop secondary bacterial infections, with higher rates observed in intensive care units [20,21]. The risk of developing bacterial co-infections remains presumably very low in non-hospitalized patients with mild disease, who represent the majority of individuals with COVID-19.

These observations suggest against the routine empirical use of antibiotics in the treatment of COVID-19 cases unless there is evidence of bacterial infection, as recommended by WHO and Indian Ministry of Health guidelines [22,23]. A few before-and-after studies have been conducted to determine the impact of COVID-19 on antibiotic use, but these were all done in high-income countries (see S1 Text and S1 and S2 Tables) [24–31].

With about 27.2 million COVID-19 cases reported as of 25 May 2021, India is among the hardest hit countries in the world [32]. In this study, we assessed the impact of the first COVID-19 epidemic wave on the consumption of antibiotics and HCQ in 2020 in India's private sector, which accounts for three-quarters of healthcare delivery and 90% of antibiotic sales in the country [33,34].

## Methods

### Study design

We conducted interrupted time series (ITS) analyses using total antibiotics, azithromycin, and HCQ sales volumes as our continuous outcome, and COVID-19 epidemic as the exposure of interest [35]. Our counterfactual (i.e., sales volumes had the pandemic not occurred) was thus the extrapolation of the pre-epidemic period. Although a formal detailed protocol was not developed, our analytic plan was designed a priori, before performing the analyses. However, our original study included sales data only up to September 2020 and was updated during the review process as more recent data became available to the study team. In the absence of a validated checklist for impact evaluation studies such as this, we followed the Strengthening the Reporting of Observational Studies in Epidemiology (STROBE) guidelines (S1 STROBE Checklist).

### Temporal data on COVID-19 in India

The first imported case of SARS-CoV-2 infection in India was identified on 30 January 2020. Until late March 2020 the number of cases detected across the country remained very low (about 0.1 per 100,000 population), although this might be partly explained by the limited number of tests being performed. In order to examine associations between drug sales volumes and COVID-19 cases, national and state-wise data regarding the monthly number of new cases detected in India were obtained from the publicly accessible online repository compiled by the Indian non-profit organization PRS Legislative Research, based on data from the Ministry of Health and Family Welfare, Government of India [32]. The monthly number of tests performed in the country was obtained from Our World in Data [36]; however, this information is not available for individual states. Projected population estimates as determined by the National Commission on Population, Ministry of Health and Family Welfare, were utilized to calculate cumulative monthly rates of new cases and tests per 100,000 population [37].

For the purpose of our regression analyses, the exposure was treated as a binary variable (pre-epidemic phase coded 0 versus epidemic phase coded 1) as detailed below.

### Antibiotic and HCQ sales data

The main outcomes of interest for this study were the sales volumes of antibiotics and HCQ in India, using data obtained from IQVIA, which is a reliable source of drug sales data [1,38]. IQVIA is a company that collects over-the-counter (OTC) and prescription-based sales data by auditing sales from a representative panel of drug stockists. The data are then extrapolated to all stockists in the country using a proprietary projection algorithm. This accounts for an estimated 95% of the total pharmaceutical market in terms of value sales combining the retail sector, hospitals, and dispensing doctors. In India, all antibiotics are included in Schedule H or H1. Schedule H is a class of prescription drugs that cannot be purchased without the prescription of a qualified doctor. For Schedule H1 drugs, in addition to having a prescription, the dispenser should record the prescriber and patient details, the drug, and the quantity dispensed and maintain the record for 3 years, and the record should be open for inspection by regulatory officials. However, OTC dispensing of antibiotics is common in India [39]. Regular monthly data points from January 2018 to December 2020 were available for the purpose of our analyses. Sales volumes were reported in standard units (SU), and 1 SU (i.e., 1 dose) was defined as a single tablet, capsule, ampoule, vial, or a 5-mL liquid preparation for oral consumption, as reported previously [38]. Information on formulation type with regard to the route of administration (oral, parenteral, topical) was also available. We further classified oral

drugs as child-appropriate formulations (CAFs) or non-CAFs based on the description of the package content (the list of formulation types considered for this purpose is provided in S3 Table), as reported previously [38]. Antibiotics were categorized according to the Anatomical Therapeutic Chemical (ATC) Index 2020 and the WHO Access, Watch, Reserve (AWaRe) framework 2019 [40,41]. The full list of drugs (intended as active molecule) included in our dataset is available in S4 Table.

## Data analysis

We performed descriptive analyses of antibiotics and HCQ sales data throughout the observation period, reporting the absolute number of doses sold along with crude percentages of each drug class relative to the total. Medians and interquartile ranges (IQRs) were also used to describe overall and stratum-specific monthly sales volumes. Descriptive analyses were also performed to explore trends up to September 2020 (peak of the epidemic wave) in selected states/territories reporting either a very high number of cases (Andhra Pradesh, Delhi, Karnataka, Maharashtra, Tamil Nadu) or a very low number of cases (Bihar, Gujarat, Madhya Pradesh, Rajasthan, West Bengal).

Next we conducted segmented regression analyses of time series data to assess how much the epidemic onset affected monthly sales volumes of (1) all antibiotics (including azithromycin), (2) azithromycin only (categorized as Schedule H), and (3) HCQ (categorized as Schedule H1 since March 2020) [35,42]. We decided a priori to exclude CAFs from these assessments as we anticipated an increase in antibiotic sales mainly among adult patients. Children constitute a small proportion of reported COVID-19 cases and are much less likely to develop symptomatic SARS-CoV-2 infection [43,44]. Furthermore, social distancing measures and school closure remained in place in most Indian states throughout the study period. As also documented in the United States [31,45], such a scenario likely plays a role in reducing the transmission of many respiratory infections that typically spread among children, leading to lower antibiotic use.

Our models for total non-CAF antibiotics and azithromycin estimated the following measures: (1) pre-epidemic trend (January 2018 to March 2020), (2) average level of change in mean monthly sales during the preventive lockdown, (3) the slope (trend) change in the outcome after the lockdown phase (i.e., from June 2020 onwards), and (4) the slope (trend) change in the outcome during the declining phase of the first epidemic wave (i.e., after September 2020) relative to the rising phase. We introduced the term "declining phase" when the analyses were updated to incorporate more recent data from October to December 2020, to better reflect the change in the epidemic trend during the study period. The model described above allowed us to account for the effect of the nationwide lockdown enforced by the Government of India between 24 March and 31 May 2020. During this time, several restrictions were imposed on the entire population, including—but not limited to—closure of schools and all nonessential services, ban on stepping out from home, and curfew. For this reason and because of the still limited circulation of the SARS-CoV-2 infection within the community, we hypothesized that antibiotic sales could have been negatively affected. A fixed effect term for the rainy season (July to October) was included in the model for antibiotics to adjust for seasonality. As this approach did not perform equally well for azithromycin sales, for this outcome we used a harmonic seasonal model to better account for seasonal changes [46], along with further adjustments for non-seasonal autocorrelation.

Given the initial recommendation for HCQ-based prophylaxis, particularly among healthcare workers [47], we expected a weaker effect of lockdown on HCQ sales and did not account for it in the model. We thus estimated the average change in level and the slope change in the

outcome assuming the start of the COVID-19 epidemic in March 2020. Autocorrelated errors were also included to correct for the remaining serial correlation in the data, whereas no adjustments for seasonality were deemed necessary. HCQ is not recommended for malaria treatment according to Indian guidelines, so no major seasonal changes are expected to occur in its use.

A detailed description of model specification and diagnostics is provided in S2 Text and S1–S3 Figs.

Descriptive analyses were performed in STATA version 16.1 (StataCorp, College Station, TX, US), and regression analyses were conducted in R (version 4.0.3).

### Ethics considerations

The institutional review boards of Washington University in St. Louis and McGill University exempted this study from ethics review as no identifiable information about living individuals was obtained (i.e., secondary use of anonymous information).

## Results

### Pattern of antibiotic and HCQ sales throughout the study period

The absolute cumulative volume of antibiotics sold in 2020 was 16,290 million doses, which is slightly lower than the 18,167 million doses and 18,002 million doses sold in the same period of 2019 and 2018, respectively (Table 1). The CAF sales volume amounted to 3,779 million doses in 2020 as opposed to 4,998 million doses in 2019 and 4,934 million doses in 2018. The proportion of non-CAF sales among total antibiotics, likely prescribed and dispensed to adolescents and adults (although pediatric and non-pediatric use are indistinguishable for injectables), increased from 72.5% (95% CI: 71.8% to 73.1%) in 2019 to 76.8% (95% CI: 76.2% to 77.5%) in 2020 (Fig 1).

The distribution of AWaRe categories remained almost stable over time except for a slight decline in the use of "discouraged" fixed-dose antibiotic combinations, which could be ascribed to a policy change introduced in September 2018 and was accompanied by an increase in the use of "access" antibiotics (Table 1; S4 and S5 Figs). The median (IQR) percentages of the different AWaRe categories relative to the total non-CAF antibiotics sold monthly throughout the entire study period (2018–2020) were as follows: "access", 43.0% (42.2%–44.4%); "watch", 36.8% (35.4%–37.6%); "reserve", 0.8% (0.7%–0.9%); and "discouraged", 18.8% (17.4%–19.7%).

The distribution of antibiotics by ATC class remained stable except for a noteworthy increase in non-CAF macrolide (J01F) sales, jumping from 947 million doses in 2018 to 1,124 million doses in 2020 (Table 1; S6 Fig). After the end of lockdown, between June and September 2020, azithromycin (J01FA10) sales were 34.4% higher than observed in the corresponding months of the previous year, followed by a decline after the peak of the first epidemic wave (Fig 1). Besides azithromycin, 2 other antibiotics, doxycycline and faropenem, that are commonly used for respiratory tract infections showed increased consumption. Monthly doxycycline (J01AA02) sales did not change much until September 2020, when a considerable peak was observed (+25.9% compared to September 2019). Faropenem (J01DI03) use has been rising constantly over the years, but a 23.4% increase was registered in September 2020 versus the year before (Fig 1). No major changes were observed in the sales volumes of other broad spectrum antibiotic classes, such as second- and third-generation cephalosporins and quinolones. Similarly, monthly sales of selected parenteral antibiotics that are typically used in inpatient care, such as carbapenems, glycopeptides, third-generation cephalosporins, and polymyxins, remained almost stable (S7 Fig).

**Table 1. Cumulative antibiotic sales volume per year (2018–2020), and distribution by AWaRe category and ATC class for formulations other than child-appropriate ones (non-CAF).**

| Category | Cumulative sales volume, in million standard units | | | | | |
| --- | --- | --- | --- | --- | --- | --- |
| | 2018 | | 2019 | | 2020 | |
| | Sales volume | Percent | Sales volume | Percent | Sales volume | Percent |
| **All antibiotics** | 18,002 | 100 | 18,167 | 100 | 16,290 | 100 |
| Non-CAFs* | 13,068 | 72.6 | 13,169 | 72.5 | 12,512 | 76.8 |
| CAFs* | 4,934 | 27.4 | 4,998 | 27.5 | 3,779 | 23.2 |
| **AWaRe category (non-CAF)**\*\* | | | | | | |
| Access | 5,479 | 41.9 | 5,656 | 42.9 | 5,621 | 44.9 |
| Watch | 4,821 | 36.9 | 4,820 | 36.6 | 4,569 | 36.5 |
| Reserve | 87 | 0.7 | 113 | 0.9 | 115 | 0.9 |
| Discouraged | 2,578 | 19.7 | 2,484 | 18.9 | 2,142 | 17.1 |
| Not included in AWaRe | 103 | 0.8 | 96 | 0.7 | 65 | 0.5 |
| **ATC class (non-CAF)**\*\* | | | | | | |
| Aminoglycosides | 240 | 1.8 | 237 | 1.8 | 184 | 1.5 |
| BL-BLI | 1,228 | 9.4 | 1,380 | 10.5 | 1,141 | 9.1 |
| Carbapenems*** | 39 | 0.3 | 46 | 0.3 | 48 | 0.4 |
| Cephalosporin-BLI | 425 | 3.3 | 477 | 3.6 | 402 | 3.2 |
| Cephalosporins (first generation) | 397 | 3.0 | 395 | 3.0 | 374 | 3.0 |
| Cephalosporins (second generation) | 232 | 1.8 | 248 | 1.9 | 216 | 1.7 |
| Cephalosporins (third generation) | 1,451 | 11.1 | 1,657 | 12.6 | 1,489 | 11.9 |
| Cephalosporins (fourth generation+) | 2 | <0.1 | 2 | <0.1 | 1 | <0.1 |
| Combinations | 2,206 | 16.9 | 2,053 | 15.6 | 1,768 | 14.1 |
| Glycopeptides | 3 | <0.1 | 4 | <0.1 | 3 | <0.1 |
| Imidazoles | 1,419 | 10.9 | 1,485 | 11.3 | 1,445 | 11.5 |
| Macrolides | 947 | 7.2 | 1,009 | 7.7 | 1,124 | 9.0 |
| Penicillins | 1,114 | 8.5 | 1,154 | 8.8 | 1,207 | 9.6 |
| Polymyxins | 2 | <0.1 | 2 | <0.1 | 1 | <0.1 |
| Quinolones | 1,718 | 13.1 | 1,664 | 12.6 | 1,546 | 12.4 |
| Sulfonamides | 329 | 2.5 | 203 | 1.5 | 321 | 2.6 |
| Tetracyclines | 869 | 6.6 | 669 | 5.1 | 611 | 4.9 |
| Other antibiotics | 446 | 3.4 | 484 | 3.7 | 467 | 3.7 |

ATC, Anatomical Therapeutic Chemical; AWaRe, Access, Watch, Reserve; BL, beta-lactam; BLI, beta-lactamase inhibitor; CAF, child-appropriate formulation.

*Percentages are calculated relative to all antibacterial drugs.

\*\*Percentages are calculated relative to non-CAFs of antibacterial drugs.

\*\*\*Including combinations of carbapenems and BLI.

Furthermore, cumulative HCQ sales (only available as non-CAF) increased by approximately 35.4% between 2019 and 2020 (from 274 million doses in 2019 to 371 million doses in 2020) (Fig 1).

## Impact of COVID-19 on antibiotic and HCQ sales

Crude monthly sales of non-CAF antibiotics increased with the number of new COVID-19 cases per 100,000 population, a pattern that is clearly observable both nationally (Fig 2) and in selected Indian states with different epidemic curves (Figs 3, S8, and S9). Rising trends are evident both in states with a high number of reported cases and in those with lower incidence. The reported number of COVID-19 cases in India remained quite low until June, reflecting

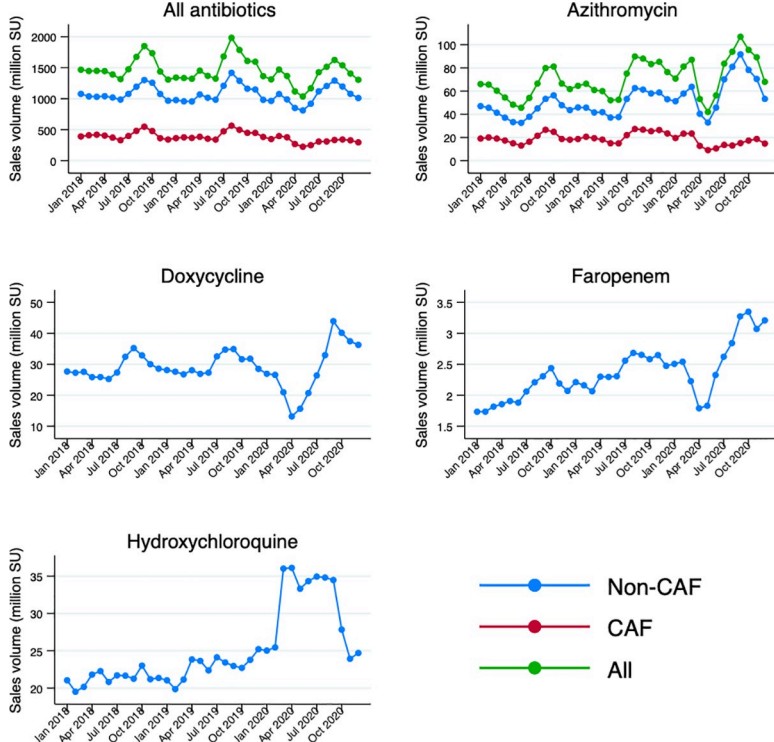

**Fig 1. Trend in sales volumes of total antibiotics, azithromycin, doxycycline, faropenem, and HCQ in India from January 2018 to December 2020.** CAF, non-CAF, and total sales are presented in the graphs, as relevant. Data on antibiotics are inclusive of azithromycin. HCQ and faropenem are only shown as non-CAFs because CAFs are not available for these drugs. As only a very small proportion of doxycycline is sold as CAF, this is omitted in the graph. CAF, child appropriate formulations; HCQ, hydroxychloroquine, SU, standard unit.

the difficulties in testing scale-up across the country, particularly during the first half of 2020 (S10 Fig).

Antibiotic sales volumes declined during the lockdown phase (April to May 2020). As estimated through segmented regression analyses (Table 2; Fig 4), non-CAF antibiotic and azithromycin sales in April 2020 decreased on average by 197.3 million doses (95% CI: −294.7 to −99.9 million; $P < 0.001$) and 11.3 million doses (95% CI: −17.6 to −5.0 million; $P < 0.001$), respectively. Moreover, we observed a monthly increase in trend after the lockdown period for both non-CAF antibiotics (+54.1 million doses [95% CI: 17.0 to 91.2 million]; $P = 0.008$) and non-CAF azithromycin (+9.5 million doses [95% CI: 6.6 to 12.3 million]; $P < 0.001$) from June to September 2020. Cumulative excess antibiotic sales from June to September 2020 amounted to 216.4 million doses (95% CI: 68.0 to 364.8 million; $P = 0.008$) for non-CAF antibiotics and 38.0 million doses (95% CI: 26.4 to 49.2 million; $P < 0.001$) for non-CAF azithromycin. The latter is equivalent to about 6.2 million azithromycin treatment courses for respiratory tract infection, considering a course to be 500 mg daily for 5 days (S2 Text). After the epidemic peak in September 2020, a declining trend in sales was observed from October to December 2020, but this was significant only for azithromycin (−20.8 million doses [95% CI: −26.93 to −14.73 million]; $P < 0.001$) (Table 2).

We also estimated a change of +11.1 million doses (95% CI: 9.2 to 13.0 million; $P < 0.001$) for HCQ sales in March 2020 (Table 3; Fig 4). After this peak, sales began declining slowly, as confirmed by the weak negative change in trend suggested by our model (−0.6 million doses

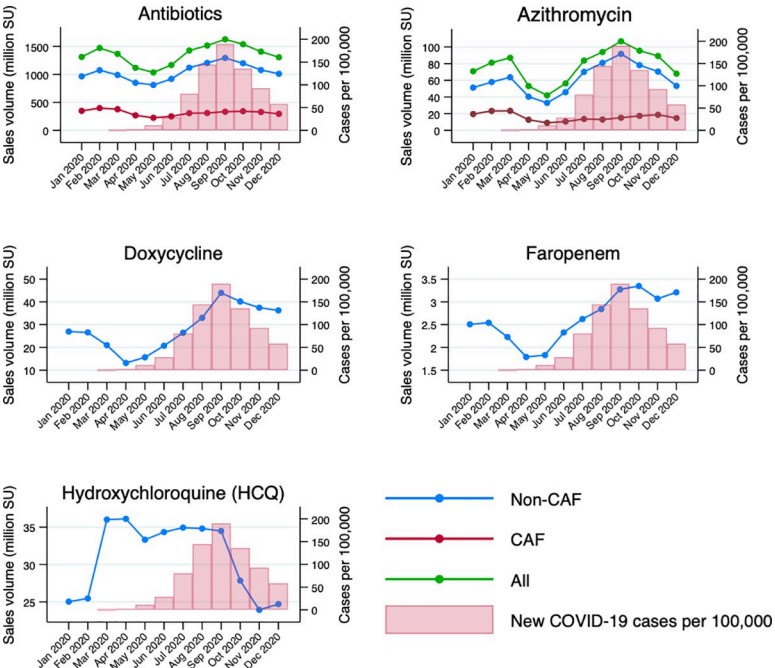

**Fig 2. Relationship between new COVID-19 cases per 100,000 (cumulative rate per month) and national sales volumes of antibiotics, azithromycin, doxycycline, faropenem, and HCQ from January to December 2020.** CAF, non-CAF, and total sales are reported. Data on antibiotics are inclusive of azithromycin. HCQ and faropenem are only shown as non-CAFs because CAFs are not available for these drugs. As only a very small proportion of doxycycline is sold as CAF, this is omitted in the graph. CAF, child-appropriate formulation; HCQ, hydroxychloroquine, SU, standard unit.

[95% CI: −1.0 to −0.1 million]; *P* = 0.010), which became more pronounced after September 2020 (−3.1 million doses [95% CI: −4.3 to −1.9 million]; *P* < 0.001).

## Discussion

We estimate that between June and September 2020, with peak epidemic activity, COVID-19 likely contributed to excess sales of 216 million doses of non-CAF antibiotics and 38 million doses of non-CAF azithromycin. The excess antibiotic sales likely resulted from the sudden surge in the number of patients seeking medical care for presumptive or confirmed COVID-19 both in the community and in the hospitals, as suggested by the abrupt increase in use of azithromycin, often prescribed for this condition. Assuming perfect adherence to the recommended dosage and duration of azithromycin treatment for respiratory tract infections not related to COVID-19 per Indian national guidelines (i.e., 500 mg daily for 5 days), 38 million excess doses from June to September 2020 corresponds to about 6.2 million treatment courses. During this period, 6 million new COVID-19 cases were reported in India across both public and private sectors, suggesting empirical use of azithromycin in the private sector in the absence of confirmed SARS-CoV-2 infection [32]. Moreover, azithromycin is often prescribed for shorter duration (e.g., 500 mg per day for 3 days) [48], potentially indicating that more people could have been treated empirically without diagnostic confirmation of SARS-CoV-2 infection. To support this, in states like Bihar, Gujarat, and West Bengal, where the number of cases is reportedly low and tests are not widely available nor accessible, azithromycin consumption has risen considerably. It should also be noted that healthcare-seeking behaviors have changed substantially during the pandemic period, with fewer people presenting to

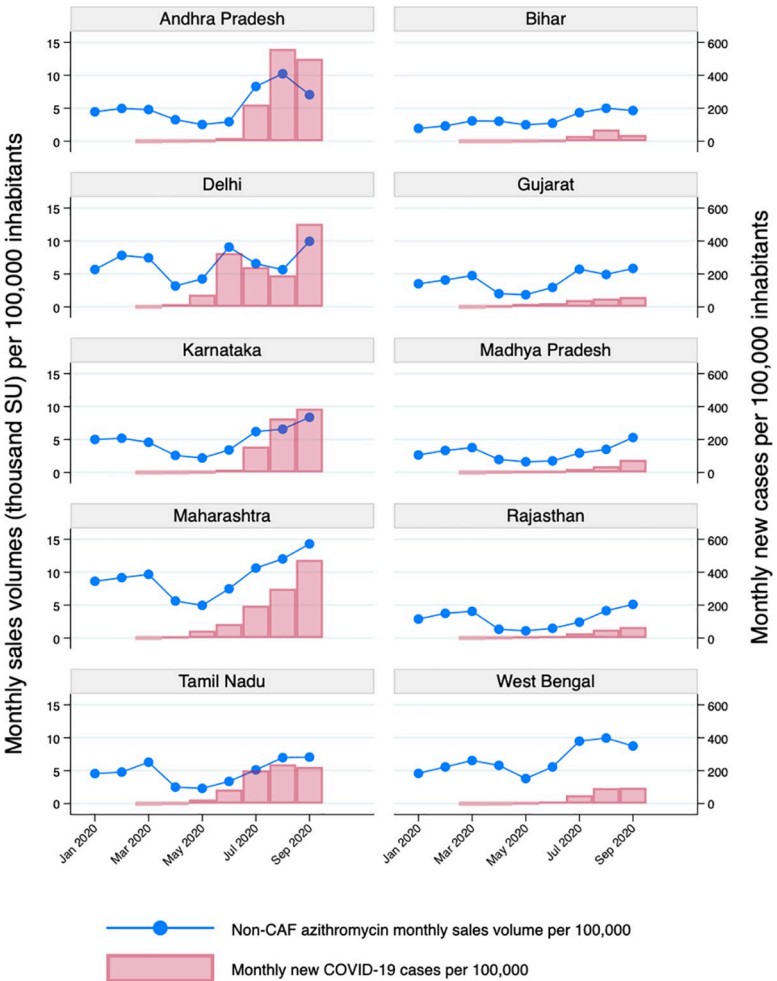

**Fig 3. Relationship between new COVID-19 cases per 100,000 (cumulative rate per month) and azithromycin sales volumes per 100,000 population (only non-child-appropriate formulations) in 10 states of India from January to September 2020.** States with the highest rates of detected COVID-19 cases are shown on the left side of the graph, whereas states with the lowest rates of detected COVID-19 cases are on the right.

healthcare facilities for conditions other than acute respiratory infections (i.e., COVID-19 suspicion). Therefore, we expect antibiotics to be less commonly prescribed for other types of illness as compared to the previous years, suggesting that the COVID-19-attributable excess sales indicated by our models might be an underestimation. On the other hand, we observed a notable reduction in CAF sales, suggesting that antibiotic use among children has declined since the start of the pandemic, which is in line with our hypotheses [31,43,45].

Notably, the massive increase in azithromycin use raises several serious concerns. First, a recent randomized controlled study investigating the effects of mass distribution of azithromycin in Nigerian children demonstrated an increase not only in macrolide resistance determinants but also non-macrolide resistance in the gut flora such as resistance to beta-lactams [49]. Multi-drug-resistant Enterobacterales, including extended spectrum beta-lactamase (ESBL)–producing strains, are highly prevalent among healthy adults in the community and could be further aggravated with this unexpected increase in azithromycin use [50]. Second, the sudden ongoing mass consumption of azithromycin has the potential to further exacerbate the selection of azithromycin-resistant typhoidal and non-typhoidal *Salmonella* strains [51]. This is of

**Table 2. Estimated change in monthly sales volume (expressed in million SU) according to adjusted segmented regression models for total antibiotics and azithromycin.**

| Measure | Sales volume in million SU | | | |
|---|---|---|---|---|
| | All antibiotics* | | Azithromycin** | |
| | Estimate (95% CI) | P value | Estimate (95% CI) | P value |
| Baseline level (January 2018) | 1,014.2 (963.4 to 1,065.1) | <0.001 | 39.4 (36.5 to 42.4) | <0.001 |
| Pre-epidemic trend (monthly change from January 2018 to March 2020) | 0.6 (−2.6 to 3.8) | 0.725 | 0.6 (0.5 to 0.8) | <0.001 |
| Average change in level during lockdown (April to May 2020) versus the pre-epidemic period | −197.3 (−294.7 to −99.9) | <0.001 | −11.3 (−17.6 to −5.0) | <0.001 |
| Change in trend after lockdown (after May 2020) | 54.1 (17.0 to 91.2) | 0.008 | 9.5 (6.6 to 12.3) | <0.001 |
| Change in trend after the epidemic peak (i.e., after September 2020) relative to the rising phase of the epidemic | −64.3 (−150.6 to 21.9) | 0.154 | −20.8 (−26.9 to −14.7) | <0.001 |

SU, standard unit. Only non-child-appropriate formulations were considered for these analyses.

*Model adjusted for seasonality using a fixed effect term indicating the rainy season.

**Harmonic seasonal model to adjust for seasonality, and autocorrelated errors to account for the remaining serial correlation in the data.

particular concern for India, where enteric fever is highly endemic, with an estimated annual incidence of 377 cases per 100,000 population, and azithromycin has been increasingly chosen for empirical treatment [52]. The recent emergence of azithromycin-resistant *Salmonella enterica* serotype Typhi (*S.* Typhi) strains in India sounds a further alarm bell [53]. Another threat coming from this unexpected increase in azithromycin use is the possible selection of

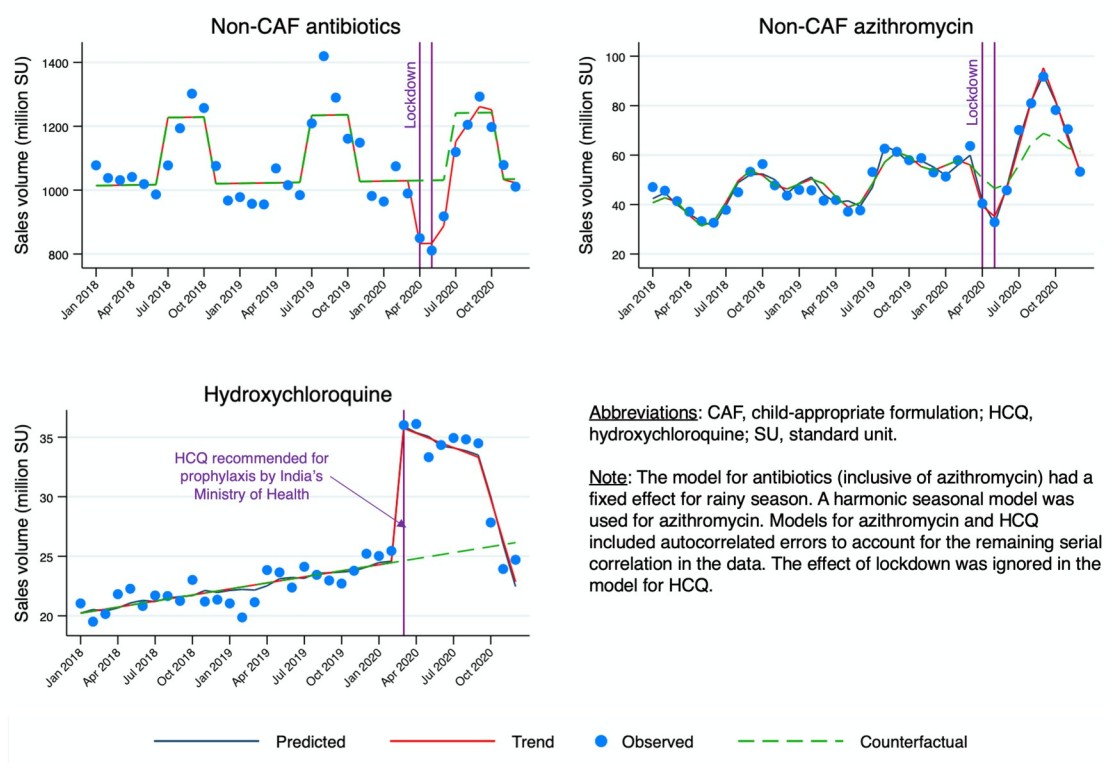

**Fig 4. Results of segmented regression analysis for monthly sales volumes of non-CAF antibiotics, azithromycin, and HCQ between January 2018 and December 2020.**

**Table 3. Estimated change in monthly sales volume (expressed in million SU) according to adjusted segmented regression models for HCQ.**

| Measure | HCQ sales volume in million SU* | |
|---|---|---|
| | Estimate (95% CI) | P value |
| Baseline level (January 2018) | 20.2 (19.3 to 21.2) | <0.001 |
| Pre-epidemic trend (monthly change from January 2018 to March 2020) | 0.2 (0.1 to 0.2) | <0.001 |
| Change in level in March 2020 versus the pre-epidemic period** | 11.1 (9.2 to 13.0) | <0.001 |
| Change in trend after March 2020*** | −0.6 (−1.01 to −0.14) | 0.010 |
| Change in trend after the epidemic peak (i.e., after September 2020) relative to the rising phase of the epidemic | −3.08 (−4.3 to −1.9) | <0.001 |

HCQ, hydroxychloroquine; SU, standard unit. Only non-child-appropriate formulations were considered for these analyses.

*Model adjusted for non-seasonal autocorrelation.

**In March 2020 (i.e., when the number of cases in India was still very low), India's Ministry of Health and Family Welfare issued a recommendation for use of HCQ in prophylaxis.

***The effect of the lockdown phase was not considered in this model based on the assumption that HCQ was predominantly used for prophylaxis among healthcare workers and thus unlikely to be negatively impacted by the lockdown as observed for antibiotics, and azithromycin in particular.

pan-oral-drug-resistant *S*. Typhi, requiring hospitalization for parenteral treatment administration [51]. Furthermore, azithromycin is currently recommended by the US Centers for Disease Control and Prevention for travelers' diarrhea in South Asia and Southeast Asia due to increasing fluoroquinolone-resistant strains among common bacterial diarrheal pathogens such as *Salmonella*, *Shigella*, and *Campylobacter* spp. [54]. The growing use of azithromycin could further jeopardize the available therapeutic choices for travelers' diarrhea. Finally, the empirical use of azithromycin for presumptive COVID-19 could lead to a progressive substitution of beta-lactam antibiotics (J01C) for any acute respiratory tract illness, aggravating the concerns regarding resistance selection.

Among other oral agents commonly used for respiratory tract infections, doxycycline and faropenem sales peaked in September 2020. Faropenem is an oral "penem" drug that has been approved in India for several clinical indications including community-acquired respiratory tract infections [55]. A recent in vitro study demonstrated cross-resistance to carbapenems among ESBL-producing *Escherichia coli* isolates [55]. The unnecessary use of faropenem could promote intestinal colonization of carbapenem-resistant Enterobacterales in a context like India where there is a high community burden of ESBL positivity. However, the decline in faropenem sales after September 2020 was not as apparent as the declines in azithromycin and doxycycline sales, indicating that faropenem may have been used for non-respiratory-tract-infection indications. Regarding HCQ, the sudden increase in consumption registered in March 2020 could be attributed to prophylaxis for healthcare workers, as initially recommended by the Ministry of Health and Family Welfare [47]. The national guidelines were subsequently revised on 27 June 2020, limiting the prescription of HCQ to moderate to severe COVID-19 cases and to patients with mild disease if immunocompromised or under 5 years old [56]. This change in recommendations is reflected in the slowly declining trend observed after the initial peak. Moreover, HCQ is unlikely to be prescribed for mild cases evaluated by primary care physicians or informal healthcare providers, who often recommend/dispense antibiotics like azithromycin but have less experience in handling HCQ-based treatment.

Additionally, in March 2020, the Indian government issued an emergency order imposing stronger restrictions on HCQ sales by including it in Schedule H1 [57]. Among the biggest threats associated with the widespread use of HCQ is the occurrence of adverse events and toxic effects, particularly when given in combination with other drugs with similar adverse effects.

There are some limitations in our study. First, IQVIA data only cover the private healthcare sector. Although this does not allow us to draw conclusions regarding the impact of the pandemic on antibiotic usage in the public sector, it should be highlighted that the private sector accounts for 75% of healthcare in India and 90% of national antibiotic sales [33,34]. Second, our data could not distinguish between inpatient and outpatient use of antibiotics, but the latter is known to be largely predominant. Third, while we applied the most appropriate techniques to adjust for seasonal variations in the outcome, the suboptimal number of pre- and post-pandemic data points available for our analyses remains a limitation in that sense. Nonetheless, our models were quite robust and fitted the data reasonably well, as the residuals of each model were behaving as white noise; yet we recommend caution in interpreting the estimated impact on HCQ sales owing to the non-stationary nature of the time series. Fourth, we did not have data in defined daily doses (DDDs); however, there is a very good correlation between DDDs and standard units when estimating antibiotic consumption per person [58]. Fifth, while data from IQVIA are widely utilized to evaluate pharmaceutical sales, it should be highlighted that the company adopts a propriety method for estimating national sales. Finally, there is a lag time between sales from stockists and purchase by the end customer at the retail pharmacy, which could not be determined through the dataset. This could be a source of measurement error in the outcome data, although the lag time was likely lower during the epidemic due to the increase in demand for antibiotics.

Our findings have important implications for antimicrobial resistance globally and even more so for low- and middle-income countries (LMICs). Like in India, the overuse of antibiotics is common in other LMICs [2,59], where similar prescribing patterns among presumptive or confirmed COVID-19 cases likely exist. The situation could be even worse in other countries like Pakistan where azithromycin is the only treatment option for *S*. Typhi infections, and an outbreak of extremely drug-resistant strains recently occurred [60]. Policy makers in India and other LMICs should recognize this substantial overuse of antibiotics induced by COVID-19. A second and substantially more devastating epidemic wave hit India from April 2021 onwards. This could result in severe antimicrobial resistance consequences if the amount of antibiotic use follows a similar pattern as in the first wave, and thus warrants reexamining the impact of the second wave on antibiotic use. Considering the ongoing trends, the very low vaccination coverage level, and the amount of time necessary to eventually vaccinate the entire population, immediate action is needed to reduce the overuse of antibiotics for COVID-19. India will need to greatly increase COVID-19 testing access to reduce empirical treatments. Issuing further restrictions on azithromycin prescription by moving it from Schedule H to H1, as done with HCQ, could potentially help limit the widespread use of this important antibiotic. Similar restrictions on azithromycin use should also be considered in other LMICs. Antimicrobial stewardship interventions have never been so critical, and mass media awareness campaigns targeting prescribers and the general public to discourage the routine use of antibiotics for COVID-19 need to be rapidly implemented in India and other LMICs.

## Supporting information

**S1 STROBE Checklist.**
(PDF)

**S1 Fig. Autocorrelation, partial autocorrelation, and distribution of residuals from model 1 for total antibiotic sales.**
(TIF)

**S2 Fig. Autocorrelation, partial autocorrelation, and distribution of residuals from model 3 for azithromycin sales.**
(TIF)

**S3 Fig. Autocorrelation, partial autocorrelation, and distribution of residuals from model 5 for hydroxychloroquine sales.**
(TIF)

**S4 Fig. Monthly sales volume of each AWaRe category in India between January 2018 and December 2020, separated for child-appropriate formulations (CAFs) and non-CAFs.**
(TIF)

**S5 Fig. Cumulative volume of antibiotics sold per year (2018–2020), stratified by AWaRe category, presented separately for child-appropriate formulations (CAFs) and non-CAFs.**
(TIF)

**S6 Fig. Cumulative volume of antibiotics sold per year (2018–2020), stratified by ATC class and presented separately for child-appropriate formulations (CAFs) and non-CAFs.**
(TIF)

**S7 Fig. Monthly sales volumes between January 2018 and December 2020 for selected antibiotic ATC classes: Parenteral carbapenems, glycopeptides, polymyxins, and parenteral third-generation cephalosporins (including those associated with a beta-lactamase inhibitor [BLI]).**
(TIF)

**S8 Fig. Relationship between new COVID-19 cases per 100,000 (cumulative rate per month) and antibiotic sales volumes per 100,000 (only non-child appropriate formulations [non-CAFs]) in 10 states of India from January to September 2020.** States with the highest rates of detected COVID-19 cases are shown on the left side of the graph, whereas states with the lowest rates of detected COVID-19 cases are on the right.
(TIF)

**S9 Fig. Relationship between new COVID-19 cases per 100,000 (cumulative rate per month) and hydroxychloroquine (HCQ) sales volumes per 100,000 (only non-child appropriate formulations [non-CAFs]) in 10 states of India from January to September 2020.** States with the highest rates of detected COVID-19 cases are shown on the left side of the graph, whereas states with the lowest rates of detected COVID-19 cases are on the right.
(TIF)

**S10 Fig. Number of SARS-CoV-2 tests performed and number of new COVID-19 cases detected each month in India per 100,000 inhabitants between January and December 2020.**
(TIF)

**S1 Table. Search strategy used in the rapid systematic review regarding the impact of the COVID-19 pandemic on antibiotic use.**
(PDF)

**S2 Table. Features and findings of studies that evaluated the impact of COVID-19 on antibiotic use.**
(PDF)

**S3 Table. List of oral formulations considered child-appropriate.**
(PDF)

**S4 Table. List of all antibiotics included in our dataset, along with ATC class, AWaRe category (2019), and Schedule H/H1.**
(PDF)

**S1 Text. Summary of the evidence regarding the impact of the COVID-19 pandemic on antibiotic use.**
(PDF)

**S2 Text. Detailed methods.**
(PDF)

## Acknowledgments

The authors gratefully acknowledge Dr. Nimalan Arinaminpathy (School of Public Health, Imperial College London, UK), Dr. Puneet Dewan (Global Health Labs, Seattle, WA, US), and Dr. Sophie Huddart (School of Medicine, University of California, San Francisco, CA, US) for providing their valuable feedback on this work.

## Author Contributions

**Conceptualization:** Giorgia Sulis, Anita Kotwani, Madhukar Pai, Sumanth Gandra.

**Data curation:** Giorgia Sulis.

**Formal analysis:** Giorgia Sulis.

**Funding acquisition:** Madhukar Pai, Sumanth Gandra.

**Investigation:** Giorgia Sulis, Sumanth Gandra.

**Methodology:** Giorgia Sulis, Brice Batomen, Sumanth Gandra.

**Project administration:** Sumanth Gandra.

**Resources:** Madhukar Pai, Sumanth Gandra.

**Software:** Giorgia Sulis.

**Supervision:** Sumanth Gandra.

**Validation:** Giorgia Sulis, Brice Batomen, Sumanth Gandra.

**Visualization:** Giorgia Sulis.

**Writing – original draft:** Giorgia Sulis.

**Writing – review & editing:** Giorgia Sulis, Brice Batomen, Anita Kotwani, Madhukar Pai, Sumanth Gandra.

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
