## [Editor Report · Decision Letter 0]

5 Feb 2021

Dear Dr Gandra, 

Thank you for submitting your manuscript entitled "Impact of COVID-19 on antibiotics and hydroxychloroquine sales in India: an interrupted time series analysis" for consideration by PLOS Medicine.

Your manuscript has now been evaluated by the PLOS Medicine editorial staff as well as by an academic editor with relevant expertise and I am writing to let you know that we would like to send your submission out for external peer review.

Kind regards,

Dr Raffaella Bosurgi

Executive Editor 

PLOS Medicine

---

## [Decision Letter · Decision Letter 1]

24 Mar 2021

Dear Dr. Gandra,

Thank you very much for submitting your manuscript "Impact of COVID-19 on antibiotics and hydroxychloroquine sales in India: an interrupted time series analysis" (PMEDICINE-D-21-00574R1) for consideration at PLOS Medicine. 

Your paper was evaluated by the editorial team and myself. It was also sent to independent reviewers, including a statistical reviewer. The reviews are appended at the bottom of this email and any accompanying reviewer attachments can be seen via the link below:

[LINK]

In light of these reviews, I am afraid that we will not be able to accept the manuscript for publication in the journal in its current form, but we would like to consider a revised version that addresses the reviewers' and editors' comments. Obviously we cannot make any decision about publication until we have seen the revised manuscript and your response, and we plan to seek re-review by one or more of the reviewers. 

We expect to receive your revised manuscript by Apr 14 2021 11:59PM. Please email us (plosmedicine@plos.org) if you have any questions or concerns.

We look forward to receiving your revised manuscript. 

Sincerely,

Dr Raffaella Bosurgi, 

Executive Editor 

PLOS Medicine

plosmedicine.org

Comments from the reviewers:

Reviewer #1: Regarding PLOS Medicine

"Impact of COVID-19 on antibiotics and hydroxychloroquine sales in India: an interrupted time series analysis"

Thank you for the opportunity to read and comment on your study. You have collected a lot of data and it is interesting to get to understand more about prescribing during the pandemic. 

My understanding of the aim is that you wanted to assess the impact of COVID-19 pandemic in India on the national consumption of antibiotics before-and-after, and to determine the impact of COVID-19 on antibiotic use in general, azithromycin and hydroxochloroquine.

Methods 

The main outcomes of interest for this study was the sales volume of antibiotics and HCQ. The drug sales data was obtained from IQVIA Inc. From your information, we understand that IQVIA Inc. collects over-the-counter (OTC) and prescription-based sales data in India through a representative panel of drug stockists and offers an overall 95% coverage of the total pharmaceutical market combining the retail sector, hospitals and dispensing doctors. I have some questions to the collection of drug data, 

1) For me this was a bit blurry - I understand that the sales data then cover hospitals and out-patients, but please explain: if you have a representative sample then the 95% coverage is actually an estimate and not true coverage?

2) Furthermore, please add whether the coverage is similar in all geographical regions

3) IQVIA Inc. collects data from drug stockists - how long lag-time would you expect from the data sold from the stockiest to the actual sale to patients (hospitals, pharmacies and dispensing doctors - and all do probably also have a stock) - do you think that could have an impact on your figures? (maybe you could touch upon this in discussion?)

Another issue that was contradictive: Under "What did the researchers do and find?" you state that "Using an interrupted time-series (ITS) design, we examined national sales volumes of total antibiotics, azithromycin alone, and hydroxychloroquine (HCQ) in India's private sector from January 2018 to September 2020". I have always understood "national sales volumes" as total sales (i.e. both private and public sector), but as you stated in discussion: "IQVIA data only cover the private healthcare sector thus…." As I read this, IQVIA Inc. data probably only cover the private sector - I suggest that this is made clearer in methods. Maybe the discussion also should be broadened a bit around this issue and whether the lack of information from the public sector matters? (Or would you say that they are similar e.g. with regard to patients drugs used, antibiotics available etc.? 

With regard to schedule H/H1, please add where azithromycin is classified. Furthermore, it would also be nice to know the prescription status of HCQ and whether that is a drug that could be sold as OTC.

 Results

In result you find that "Antibiotic sales volumes declined in April and May 2020" and then you have added "likely due to the very limited mobility allowed during the lockdown phase" But could another explanation be because it was less access to antibiotics due to mobility - or because it was less need - i.e. lower infection pressure? And further you say "Moreover, we observed a monthly increase in trend after the lockdown period for non-CAF antibiotics" I suggest to addressed this further in discussion. 

Discussion

Regarding "HCQ, the sudden increase in consumption registered in March 2020 could be attributed to prophylaxis for healthcare workers as initially recommended by the Ministry of Health". I agree, increased use was seen in many countries in the first days of the pandemic, however HCQ was also given as treatment to COVID-19 patients.

Under limitations

With regard to the coverage of data (see also Q in methods); private versus public sector "First, IQVIA data only cover the private healthcare sector thus potentially underestimating the excess use of antibiotics and HCQ due to COVID-19". Could it be the other way around - that there is excess use in private sector and not in the public sector? 

With regard to inpatient versus outpatient use: "Second, our data could not distinguish between inpatient and outpatient use of antibiotics……". Somehow you could, as you state in results; "Similarly, monthly sales of selected parenteral antibiotics that are typically used in inpatient care such as carbapenems, glycopeptides, third generation cephalosporins and polymyxins, has remained almost stable". Don't this indicate that the changes in antibiotics could be more due to the use in outpatients than the use in hospitals?

With regard to seasonal variations in the outcome, see comments in Text S2

With regard to other possible treatments for COVID, you mention that ivermectin is also used off-label for COVID-19 treatment in India, but what about other treatment regimens that have been discussed during the pandemic e.g. corticosteroids, some antivirals etc?

Then you mention DDDs: " Finally, we did not have data in daily defined doses (DDDs); however, the antibiotic consumption data in standard units correlates very well with DDDs" But your reference (no. 46) refers to DOT and not to DDDs. 

Text S2: Detailed methods, model 4

Here you claim that HCQ is predominantly used as an immunomodulator and most commonly for non-infectious conditions, hence adjustment for seasonality was deemed unnecessary" 

However, HCQ is also indicated for Malaria (both prophylaxis and therapy) - and those are approved indications among others both in Europe and in US (FDA ), therefore it is strange if you do not have those indications in India. Moreover, according to an article about Seasonal variations in incidence of severe and complicated malaria in central India (Indian J Med Sci. 2001 Jan;55(1):43-6.) claims that the maximum prevalence of malaria in most parts of India is from July to November months, which is the malaria season in India and are among the months you address in the study. It would be nice to address this point in discussion 

Table S4: List of all antimicrobials included in our dataset, along with AWaRe (2019) and ATC categories (2020).

In the column for ATC you have grouped antibacterials, but not necessarily according to ATC. If you want to name the column ATC you should use the ATC code in addition to the antibiotic group, e.g. J01G Aminoglycosides. I also see that you diverge from the ATC groups for some of the substances e.g. tigecycline is grouped under tetracyclins (J01AA) in the ATC-system. It is of course OK to group the antibiotics the way you decide, and it is very clear in the Table how you group them, but be aware that it is not ATC grouping.

For international readers it would have been nice to have an extra column for which antibiotics are schedule H, H1 and OTC

Then, in the footnote you have mentioned : "CQ, chloroquine; HCQ, hydroxychloroquine" I could not find that you had used the abbreviations in the Table? 

Reviewer #2: 

Thanks for the opportunity to review your manuscript. My role is as a statistical reviewer, so my comments and questions are focused on the data and analysis. This manuscript presents an interrupted time-series analysis of national-level sales of antibiotics (AZM and the one we have heard so much about in the last 12 months, HCQ). I've put general comments/queries first and followed this with questions about specific sections of the manuscript.

The manuscript is good quality, it's well-written and clear and the approach to analysis is overall good (the details in the Supplementary appendix weren't just helpful for the review but were interesting to read as well). 

The supplementary appendix was a useful addition for my review - the detailed methods you provided were very helpful in my review and answered several questions I had from reading the main manuscript. The final models selected after observing autocorrelation in the original formulation are appropriate and the limitations with the model for HCQ are acknowledged.

My main concern is about the data used in the analysis. The data is sourced from a private company that estimates sales through sampling over the counter sales and prescription records. This is briefly discussed in S2 Text - that there is a risk that the approach used to estimate national sales becomes biased during the COVID-19 period of 2020. The references (1, 25) provided are for two published studies with international comparisons (which include India) using the same database. I spent some time looking through IQVIA's website, but I didn't find any more detailed information. I was looking for the specific methodology used to estimate the total sales information (including how this was done in India) and any quality assessments of their methodology. The two publications provided to point to this being a legitimate source of data but for the review I would like to see the methodology used to estimate total sales so that I can make a complete and robust assessment of this work. I don't harbor any suspicions here but after the retraction of the work using the Surgisphere dataset last year in The Lancet I would like to be able to fully recommend your work with no doubts about the data used in the analysis. 

P9. The association between sales volume and new COVID-19 cases was analysed with Pearson's correlation. There is a risk with this approach of an inflated correlation coefficient due to common underlying trends captured as an 'association'. There are a variety of approaches to dealing with this e.g. in the environmental epidemiology context I have used GAMs with a non-linear time effect to detrend and get a less biased estimate of association. I would recommend to either use a more appropriate approach (there are several to choose from) or to leave this out and present the summary figures which show a clear pattern of association as is.

In the figures (particularly Fig 1) the x-axis is quite noisy with the month and year displayed for each data point - this could be refined (e.g. year not displayed in each label, not every month labelled on the axis) to be clearer.

In Figure 2, I was unclear about the rate. Is this the rate of new infections per 100,00 in each month, or a cumulative rate of all COVID19 infections to the end of each months?

Text S2. The method used is described as 'generalized linear models with least-squares', I think this would be more accurately called ''general linear model' if least squares where used with a normal distribution.

Reviewer #3: This is important data to show the wider impact of COVID-19 on antibiotic use. I have some clarification comments and suggestions for strengthening the introduction and discussion. For example, I think more is needed on the background on azithromycin and HCQ use, as well as what a "lockdown" was in India. There is also some subtlety that could be included around the discussion about when and why antibiotic use might change with COVID-19 (e.g. direct and indirect effects). I cannot assess the time series analysis in detail but I think more justification of the choices made is needed. 

Minor comments

Introduction

- COVID-19 is the disease with symptoms not "infection designated as coronavirus disease". Unclear please change

- Why were people using azithromycin and HCQ? More information on the early evidence on this regimen and subsequent rebuttal

- More subtlety is needed around why a patient might receive an antibiotic: the symptoms of hospitalised COVID-19 look like a bacterial pneumonia. Similarly, in the community, it is unlikely that people are taking an antibiotic because they think they have a bacterial infection but more because there are no treatments for COVID-19 and they want to take something

- What are azithromycin / HCQ used for in India "normally"? (i.e. pre-COVID-19?)

Methods

- Why is the information on schedule H / H1 in there? 

- Why does social distancing / school closures affect why CAF formulas were not included? Suggest to cut this sentence (middle page 9), unless they could also assess the impact of such NPIs on the number of respiratory infections and hence the reduction in CAF antibiotic use. 

- The recommendation for HCQ-based prophylaxis should be included in the introduction - was this just for India? 

- (Top of page 10) If HCQ is not recommended for malaria why is it a concern to be used more for COVID-19? More background on this needed in the introduction. 

- What does it mean " this approach did not perform equally well for azithromycin"? I think the model needs to be the same for all or at least all the same model types tested for each data set. 

Results

- How are the confidence intervals generated around the proportion of non-CAF? Why do you have confidence intervals? Is it not just a single proportion? 

- How did AWaRE classifications vary in the COVID period?

- Is azithromycin a macrolide? Can you explicitly say that this could have caused the jump? (bottom of page 11). 

- Figure 2: the trend is not completely obvious to me and why is there no trend with HCQ? Can you discuss /explain the non-trends in this graph? 

- Figure 3: the correlation / trend is clearer here: did you analyse this statistically? 

- It's not completely clear to my why lockdown would affect antibiotic usage - can explain what "lockdown" meant in the Indian setting and this link in the background. 

- How did doxycycline levels compare to September 2018? Why is this and farpenem singled out for analysis? 

- Why was there a change in September 2020? 

- In April 2020 non-CAF and azithromycin decreased vs 2018/19 or the start of 2020? 

Discussion 

- Is azithromycin really often prescribe for this condition? In India private / OTC settings? 

- Is there a potential that other diseases, not being treated in hospital settings, could explain the increase in some antibiotic usage? 

- In line with the above why might you expect the number for COVID-19 to be an underestimate? Surely if not attending healthcare facilities they may self-medicate? 

- Could some of this background on azithromycin use go into the introduction to justify the focus and concern? 

- What proportion of the healthcare prescribing in India is not in the private sector? (to assess the limitations)

[LINK]

---

## [Decision Letter · Decision Letter 2]

26 May 2021

Dear Dr. Gandra,

Thank you very much for re-submitting your manuscript "Impact of COVID-19 on antibiotics and hydroxychloroquine sales in India: an interrupted time series analysis" (PMEDICINE-D-21-00574R2) for consideration at PLOS Medicine.

I have discussed the paper with editorial colleagues and it was also seen again by two reviewers. I am pleased to tell you that, provided the remaining editorial and production issues are dealt with, we expect to be able to accept the paper for publication in the journal.

The issues that need to be addressed are listed at the end of this email. Any accompanying reviewer attachments can be seen via the link below. Please take these into account before resubmitting your manuscript:

[LINK]

Please let me know if you have any questions, and we look forward to receiving the revised manuscript.   

Sincerely,

Richard Turner, PhD

rturner@plos.org

Requests from Editors:

Please adapt the title to "Sales of antibiotics and hydroxychloroquine in India during the COVID-19 epidemic: An interrupted time-series analysis" or similar.

At line 32, please make that "the private sector".

At line 47, would "... first epidemic wave ..." be appropriate?

In the abstract and throughout the paper, please quote p values alongside 95% CI where available. 

Please add a new final sentence to the "Methods and findings" subsection of your abstract, which should begin "Study limitations include ..." or similar and should quote 2-3 of the main study limitations. 

At line 82, please make that "... likely to have occurred in other LMICs ...". 

Early in your Methods section, please state whether or not the study had a protocol or prespecified analysis plan, and if so attach the relevant document(s) as a supplementary file, referred to in the text. 

Please highlight non-prespecified analyses. 

Please remove the information on funding and competing interests from the end of the main text. In the event of publication, this information will appear in the article metadata via entries in the submission form.

Throughout the text, please remove spaces from within the reference call-outs (e.g., "... post-exposure prophylaxis [13,14].").

In the reference list, please use the abbreviation "PLoS ONE".

Noting references 36 and 57, can some additional access information, or a URL, be added?

Please add "[preprint]" to reference 53 and any other cited preprints, and correct the mis-formatting.

Please add a completed checklist for the most appropriate reporting guideline, e.g., STROBE, as a supplementary document. This should be labelled "S1_STROBE_Checklist" or similar and referred to as such in your Methods section. 

In the checklist, please refer to individual items by section (e.g., "Methods") and paragraph number, not line or page numbers (as the latter generally change in the event of publication).

Comments from Reviewers:

*** Reviewer #2: 

Thanks for the revised manuscript and response to my original queries. Figures 1 and 2 are both improved and clear to read. The additional information on IQVIA was helpful - the IQVIA pharmacy data does seem to be widely used and legitimate. 

The limitation with the data is that without information about the proprietary method that is used to estimate national data from the sample panel of sellers there is the possibility that this method becomes biased under unusual conditions, e.g. COVID-19. This is acknowledged with respect to changes in lag-time, I think that a general acknowledgement of the limitation of using a propriety estimation method, along with the specific issue of lag time in the discussion would be a sufficient clarification of the issue.

*** Reviewer #3: 

[supportive report received]

***

[LINK]

---

## [Editor Report · Decision Letter 3]

30 May 2021

Dear Dr Gandra, 

On behalf of my colleagues and the Academic Editor, Dr Knight, I am pleased to inform you that we have agreed to publish your manuscript "Sales of antibiotics and hydroxychloroquine in India during the COVID-19 epidemic: an interrupted time series analysis" (PMEDICINE-D-21-00574R3) in PLOS Medicine.

Prior to final acceptance, please amend the typo in the file name for the STROBE checklist.

PRESS

Sincerely, 

Richard Turner, PhD 

rturner@plos.org